# Glyphosate-Based Herbicide Formulations and Their Relevant Active Ingredients Affect Soil Springtails Even Five Months after Application

**Anna Altmanninger** [1], **Verena Brandmaier** [1], **Bernhard Spangl** [2], **Edith Gruber** [1], **Eszter Takács** [3], **Mária Mörtl** [3], **Szandra Klátyik** [3], **András Székács** [3] and **Johann G. Zaller** [1,*]

1  Department of Integrative Biology and Biodiversity Research, Institute of Zoology, University of Natural Resources and Life Sciences Vienna (BOKU), Gregor Mendel Straße 33, 1180 Vienna, Austria; anna.altmanninger@haup.ac.at (A.A.); verena.brandmaier@gmx.at (V.B.); edith.gruber@boku.ac.at (E.G.)

2  Department of Landscape, Spatial and Infrastructure Science, Institute of Statistics, University of Natural Resources and Life Sciences Vienna (BOKU), Peter-Jordan-Straße 82, 1190 Vienna, Austria; bernhard.spangl@boku.ac.at

3  Agro-Environmental Research Centre, Institute of Environmental Sciences, Hungarian University of Agriculture and Life Sciences, Herman Ottó út 15, H-1022 Budapest, Hungary; takacs.eszter84@uni-mate.hu (E.T.); mortl.maria@uni-mate.hu (M.M.); klatyik.szandra@uni-mate.hu (S.K.); szekacs.andras@uni-mate.hu (A.S.)

*  Correspondence: johann.zaller@boku.ac.at; Tel.: +43-1-487654-83318

**Abstract:** Glyphosate is the most widely used active ingredient (AI) in glyphosate-based herbicides (GBHs) worldwide and is also known to affect a variety of soil organisms. However, we know little about how the effects of glyphosate AIs differ from those of GBHs that also contain so-called inert co-formulants. We conducted a greenhouse experiment using the model cover crop white mustard (*Sinapis alba*) to investigate the effects of three GBHs (Roundup PowerFlex, Roundup LB Plus, and Touchdown Quattro) and their respective glyphosate AIs (glyphosate potassium, isopropylamine, and diammonium salt) on epedaphic springtails (*Sminthurinus niger*; Collembola) activity in soils with low (3.0%) or high (4.1%) organic matter content (SOM). Springtail activity was assessed using pitfall traps. Most GBHs and AIs reduced springtail activity compared to mechanical removal of mustard in the short-term and even up to 5 months after application. GBHs and AIs differed considerably in their effects on springtail activity, and effects were modified by SOM content. Our results highlight the need to (i) distinguish between the effects of glyphosate AIs and commercial GBH formulations, (ii) disclose all ingredients of GBHs, as co-formulants also affect non-target organisms, and (iii) include soil properties in ecotoxicological risk assessments for soil organisms to better characterize the situation in the field.

**Keywords:** agrochemicals; chemical weed control; conventional agriculture; non-target effects; soil fauna; ecosystem function; soil ecology

## 1. Introduction

Glyphosate is the most commonly used broad-spectrum active ingredient (AI) in many herbicides worldwide [1]. Glyphosate-based herbicides (GBHs) are mainly used for post-emergent weed control in a variety of annual and perennial crops, pre-harvest desiccation [2], cover crop removal [3], and as a common practice in minimum tillage, no-till, or similar conservation agricultural practices [4]. Outside of agriculture, it is used along roadsides, railroad tracks, and in private gardens [1,5].

GBH formulations contain about 35–75% of the AI glyphosate, possibly other AIs, and so-called inert co-formulants [6]. Glyphosate AI exists in various forms as either glyphosate acid or glyphosate salt [7]. There are about 2000 differently formulated GBHs in use worldwide [8].

Co-formulants facilitate the permeation of plant cells, preserve the active ingredient from degradation, increase its solubility and half-life, and enhance its activity. However, the amount and type of co-formulants added to GBHs are guarded as trade secrets [9]. The manufacturing companies refer to these co-formulants as inert and claim that they are not involved in the toxicity of the herbicides. However, several studies have shown that co-formulants may be more toxic than the AIs themselves [6,10–12]. Formulations contain a variation of different adjuvants that are rarely tested as a mixture, resulting in completely unknown cocktails and synergetic effects on the environment and human health [13]. The GBH formulation Roundup was found to have 125-fold increased cytotoxic effects than the active ingredient glyphosate alone [10]. Glyphosate and its formulations have been shown to affect animal behaviour in various taxa [8,14].

Glyphosate enters the soil when sprayed on weeds, through runoff, soil erosion, air, root exudates, or decomposition of sprayed plant residues. Consequently, epigeic or endogeic soil fauna may come into contact with glyphosate [15]. The fate of glyphosate in soil is influenced by sorption and desorption processes, which are modified by soil minerals and soil organic matter content (SOM) [16]. In general, in soils with a higher SOM content, aluminium or iron oxides can bind glyphosate to a greater extent [1,17]. Glyphosate dissolved in soil solution is available to microorganisms and is degraded more rapidly [18] and may even stimulate some soil microorganisms [19]. Glyphosate bio-availability is also influenced by soil properties such as SOM [20], pH, salinity, and nutrient content [21]. The persistence of glyphosate in soil varies considerably from low to very high with a half-life ranging from 2.8 to 500.3 days [2]. Consequently, glyphosate and its metabolites have been found in soils throughout Europe and have been shown to enter surface waters [22] or even via ambient air [23].

Soil fauna has been shown to be directly and indirectly affected by GBH or glyphosate AI [8,24,25]. Among the most important members of soil mesofauna in temperate soils and indicators of soil quality and sustainable land use are springtails (collembola). Springtails are 0.2 to 9 mm small wingless hexapods and are among the most abundant soil arthropods with 6500 species worldwide, reaching densities up to 100,000 individuals m$^{-2}$ [26–28]. They contribute to nutrient cycling, decomposition of dead organic matter, and the construction of microstructures, and are an important food source for many other species [29,30].

Springtails are often used as surrogate species in ecotoxicological studies [31,32]. However, standardized toxicity tests for springtails, in which they are exposed to the substances for up to four weeks [21], rarely consider long-term effects. GBHs have been shown to affect soil microorganisms even 11 months after application and also affect the following crops [19,33] and affect crop health and nutrient [34], plant defence and species interactions [35]. These changes are expected to have ecosystem and even evolutionary consequences for both microbes and hosts [36]. Since springtails are among the most sensitive groups of soil fauna to a variety of pesticides, corresponding effects are expected for GBHs or their AIs [37]. Overall, mixtures of chemicals have been shown to have synergistic effects and increase toxicity to both reproduction and survival of springtails when applied in combination, as opposed to when applied individually [38]. Both GBHs and AIs increased the surface activity of springtails compared to control pots, and the activity of springtails was higher under GBHs than under corresponding AIs, which was interpreted as a stress response [39].

To our knowledge, only one study to date has investigated the effects of GBHs and AIs interacting with different soil types on springtail activity [39]. The research questions of the current study were (i) do commercial herbicide formulations (Roundup Power-Flex, Roundup LB Plus, and Touchdown Quattro) and their respective AIs (glyphosate potassium, isopropylamine, and diammonium salts, respectively) differ in their effects on springtail activity, and (ii) to what extent influence different soil types the effects of GBHs and AIs on springtails. We investigated both the long-term legacy effects when weed control was applied five months ago and the short-term effects after an additional application of the same GBH/AI treatments. Based on previous studies, we expected that (i) the appli-

cation of single active ingredients (AI) glyphosate, as well as glyphosate-based herbicide formulations (GBHs), would reduce springtail activity compared to untreated pots, (ii) springtails would have increased surface activity in soil with higher SOM, and (iii) there would be no legacy effects due to short half-life of AIs, but this is unclear for GBHs.

## 2. Materials and Methods

This study was conducted between October and December 2018 and investigated the short- and long-term effects of weed control using three glyphosate AIs, and three GBHs; the control treatment was hand weeding. We used the soil of a greenhouse pot experiment conducted between April and July 2018 using the same weed control methods [39,40]. After completion of the previous experiment, the soil was stored in a greenhouse. Instead of defaunating the soil by freezing, heating, microwaving or with chemicals, we allowed it to dry out completely during the summer months to deplete soil fauna. We assumed that potential differences in soil fauna of the different soil types were equalized after this treatment.

The current experiment consisted of two parts. First, the long-term legacy effects of the GBH/AI application on 13 June 2018 were studied for up to 150 days. Second, the short-term treatment effects were studied over 35 days after another application of the same treatments on 13 November 2018.

### 2.1. Experimental Setup

The experiment was conducted from 3 October 2018 to 17 December 2018 (76 days in total) at the research greenhouse of the University of Natural Resources and Life Sciences (BOKU), Vienna, Austria. The experimental setup consisted of 70 plastic plant pots (height 23 cm, diameter 31 cm, volume of 17.4 l) filled with 17 kg of topsoil from arable fields at the BOKU research farm in Groß-Enzersdorf near Vienna. We used soils with different soil organic matter (SOM) contents and filled 35 pots with soil containing 4.1% soil organic matter (=high SOM) and the other 35 pots with soil containing 3.0% SOM (=low SOM). The low SOM soil was conventionally farmed in a crop rotation with common pesticide treatments, but no AIs or GBHs were applied in the last three years. The high SOM soil had been managed organically for the past two decades and had not been treated with herbicides or other synthetic pesticides since that time. Both soils had a pH of 7.7. The high SOM soil had a phosphorus (P) = 113 mg kg$^{-1}$, potassium (K) = 234 mg kg$^{-1}$; the low SOM soil had a P = 73 mg kg$^{-1}$, K = 140 mg kg$^{-1}$. Soils were classified as calcareous Chernozems.

White mustard (*Sinapis alba*) was sown in each pot as a model crop, as it is a typical winter cover crop in Austria that is sprayed with GBH in spring after mild winters that do not kill the cover crop. We used certified organic seed material (Reinsaat, St. Leonhard am Hornerwald, Austria). The recommended seeding density was 2.5 g m$^{-2}$. Accordingly, 28 seeds pot$^{-1}$ were sown at a 2 cm depth in a uniform pattern on 11 October. The pots were irrigated regularly in response to soil dryness. Irrigation was equivalent to 70.7 mm of rain throughout the experiment.

A two-factorial design with the factors Treatment, SOM was established:

- Factor Treatment (7 levels):
  - GBHs: one-time application of Roundup PowerFlex (PF), Touchdown Quattro (TQ), and Roundup LB Plus (LB).
  - AIs: one-time application of potassium salt (po; AI of PF), diammonium salt (am; AI of TQ), and isopropylamine salt (is; AI of LB).
  - Mechanical removal of mustard (CO).

- Factor SOM (2 levels):
  - low (3.0%).
  - high (4.1%).

Each combination was replicated five times, resulting in 7 treatments × 2 SOMs × 5 replications = 70 pots. More details on the treatment applications follow below.

*2.2. Springtails*

On 12 October, one day after seeding, 200 individuals of the Collembola species *Sinella tenebricosa* (family Entomobryidae) were introduced in each pot. Springtails were purchased from various stores in Austria (Megazoo, Brunn am Gebirge; Hornbach Garden Centre; Brunn am Gebirge; Zoo Austria, Vienna; Terra Reptilia, Vienna, Austria). Microscopic identification confirmed that the springtails from the different vendors belonged to the species, *S. tenebricosa* [41]. Springtails were counted on black cardboard and separated with an aspirator under a magnifying glass.

To prevent springtails from escaping the pots, a 25 cm high barrier of transparent plastic film, commonly used for polytunnels, was attached to the top edge of the experimental units. The upper inner edges of the plastic film were additionally coated with soft soap.

*2.3. Herbicide Applications*

Thirty-three days after seeding, the GBHs and AIs were applied using a water sprayer with an attached fine mist spray nozzle on 13 November 2018. Both GBHs and AIs were applied to the mustard plants according to label recommendations (Table 1). Control pots were sprayed with tap water at the rate required for GBHs/AIs. Mustard plants in the control pots were uprooted on 16 November 2018 and their residues remained on the soil surface. The experimental pots only contained mustard plants as model weeds, but no other plant species. As our aim was to compare the effects of the commercial GBHs with their respective glyphosate AI content, we evaluated the inherent AI concentrations of these GBHs and not the same concentrations for all substances.

**Table 1.** Recommended and applied herbicide amounts of the GBHs Roundup PowerFlex (PF), Roundup LB Plus (LB), and Touchdown Quattro (TQ), and their AIs potassium salt (po), isopropylamine salt (is), and diammonium salt (am). Pot area 0.075 m$^2$.

| GBH | AI | Recommend Application (L ha$^{-1}$) | AI Content (g L$^{-1}$) | GBH (mL pot$^{-1}$) | AI (g pot$^{-1}$) |
|-----|-----|-----|-----|-----|-----|
| PF | po | 3.75 | 588 | 0.0281 | 0.0165 |
| LB | is | 5.0 | 360 | 0.0375 | 0.0135 |
| TQ | am | 5.0 | 360 | 0.0375 | 0.0135 |

The GBHs consisted of Roundup PowerFlex (PF) (AI 588 g L$^{-1}$), Roundup LB Plus (LB) (AI 360 g L$^{-1}$) (both Bayer Agrar Austria; Vienna, Austria) and Touchdown Quattro (TQ) (AI 360 g L$^{-1}$) (Syngenta Agro GmbH; Vienna, Austria). The GBHs Roundup PowerFlex and Roundup LB Plus were purchased at Raiffeisen-Lagerhaus GmbH at Bruck an der Leitha, Lower Austria. Touchdown Quattro was ordered online from vmd-drogerie, Veseli nad Moravou, Czech Republic.

The AIs consisted of various salts of glyphosate (N-phosphonomethyl-glycine): potassium salt (AI in Roundup PowerFlex), isopropylamine salt (AI in Roundup LB Plus), and diammonium salt (AI in Touchdown Quattro). The AIs were obtained from commercial sources or synthesized at the Agro-Environmental Research Institute of the National Agricultural Research and Innovation Centre, Budapest, Hungary. Glyphosate isopropylamine salt was purchased from Toronto Research Chemicals (North York, ON, Canada), while the potassium and diammonium salts were prepared from glyphosate purchased from Sigma-Aldrich, Hungary (Budapest, Hungary). Thus, 1.66 g (9.82 mmol) of glyphosate was gradually added under continuous stirring to a cooled 0.84 mL aliquot of a 45% (*wt wt*$^{-1}$) aqueous potassium hydroxide solution. The mixture was stirred overnight at 4 °C, the resultant precipitation was filtered and lyophilized to yield 1.04 g (5.02 mmol, 51.1%) of glyphosate potassium salt. Similarly, 1.66 g (9.82 mmol) of glyphosate was gradually added under continuous stirring to a cooled 1.33 mL aliquot of a 28% (*wt wt*$^{-1}$) aqueous diammonium hydroxide solution. The mixture was stirred overnight at 4 °C, the resultant precipitation was filtered and lyophilized to yield 1.01 g (4.97 mmol, 50.6%) of glyphosate

diammonium salt. The solubility of glyphosate diammonium salt was 144 g L$^{-1}$ [42], and monopotassium salt was 923.3 g L$^{-1}$ [43].

## 2.4. Measurements

Springtail activity was determined using pitfall traps [27,44]. Five Eppendorf tubes (volume 1.5 mL, diameter 1 cm) per pot were used as pitfall traps, which were randomly inserted into each pot. They were filled with 1 mL of ethylene glycol and one drop of odourless detergent per 100 mL to reduce surface tension to prevent the springtails from escaping. Thus, 350 traps per sampling date were buried to be at level with the soil surface in a 20-cm-circle in each pot. On 18 October, six days after the springtail addition, the pitfall traps were placed for the first time. These were inserted at nine sampling intervals usually in the same locations as before. In the first sampling, they were removed after 48 h, and in the second after 72 h. Each of the remaining seven samplings remained in the soil for 96 h. The collected springtails were counted under a light microscope.

Soil measurements were taken ten times in all pots during the experiment using a portable time domain reflectometer system (TDR; HD2, TRIME®-PICO 64/32, IMKO Micro-modultechnik GmbH, Ettlingen, Germany). This is a fast method for accurate measurement of soil moisture (%), temperature (°C), and electrical conductivity (dS m$^{-1}$). These three parameters were measured five times before and five times after treatment. The TDR measuring fork was placed in the centre of the pots for the measurements. After the first time, sticks were inserted into the holes to keep the device inserted in the same place. The air temperature was measured automatically with a data logger installed in the greenhouse (MTV-model, HortiMax growing solutions, Hortisystems, Pulborough, UK). The mean air temperature throughout the entire experiment was 17 ± 2.4 °C. Shading and ventilation in the greenhouse were set to automatic mode and the heating was switched off.

The height of the mustard plants was measured on the day of herbicide application. For this, five randomly selected plants per pot were measured with a ruler from the soil surface to the top of the plant and the heights per pot were averaged.

## 2.5. Statistical Analysis

The data analysis was performed using R version 3.6.3 [45], with the significance level ($\alpha$) set at 0.05. The main factors "treatment" with seven levels (co, PF, po, LB, is, TQ, am) and "SOM" with two levels (low and high) and their interaction were included in all statistical models.

Generalized linear models (GLMs) with a negative binomial distribution (package "MASS") and "log" links were created to model the effect of parameters on springtail surface activity. Four different models were created to test for legacy effects on springtail activity and cumulative springtail activity of the treatment 150 days prior, and the short-term effects of treatment on springtail activity and cumulative activity over 34 days. The test predictor "day of experiment" was included in the models as a numerical predictor.

Linear models (LMs) were fitted to TDR measurements. Soil temperature, moisture and electrical conductivity were set as predictors in all models. The residuals of the models were tested for normal distribution by plotting them. Two models were fitted to each TDR value, testing the five measurements before and after treatment, respectively, including the "day of experiment" as a predictor.

To test for significant effects of the predictor variables on the response variables, type II analyses of variance were performed for all models using the "Anova" function in the "car" package [46]. Differences between the seven levels of "treatment" were assessed using a general linear hypothesis test "glht" in the package "multcomp" [47] with pairwise Tukey comparisons. This function tests the means of the chosen predictor of all levels and adjusts the *p*-values using the single-step method. All *p*-values reported for differences between the treatment levels are adjusted *p*-values.

## 3. Results

### 3.1. Long-Term Legacy Effects on Springtail Activity

During the seven sampling dates, we caught a total of 3560 springtails, out of 14,000 specimens added. Ninety-five percent of them (3374 specimens) were identified as *Sminthurinus niger* (Symphypleona), and 186 specimens belonged to other families. Interestingly, the species originally released in the pots, *Sinella tenebricosa,* were not captured by the pitfall traps. Therefore, we used the data from *S. niger* for our evaluation and justified this by the fact that the soil was well-homogenized and treated equally before it was used for this experiment.

Mean springtail activity was highest in control pots, followed by AI and GBH pots (Figure 1, Table 2). We found significant legacy effects of the factor treatment, SOM, and a treatment × SOM interaction (Figure 1, Table 2). Average springtail activity across all treatments was greater in high SOM pots (4.6 ± 0.7 individuals pot$^{-1}$, mean ± SE) than in low SOM pots (2.9 ± 0.5 ind. pot$^{-1}$).

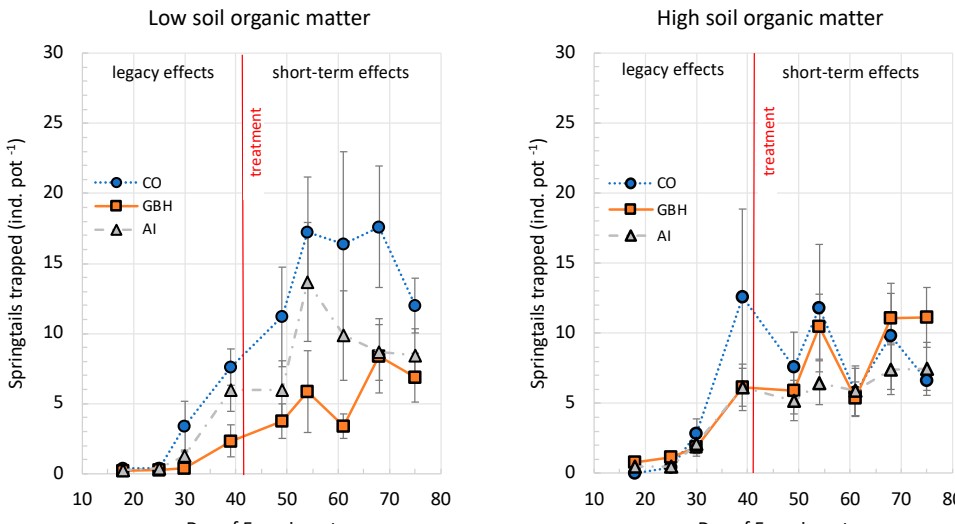

**Figure 1.** Springtail activity under low (3.0%) and high (4.1%) soil organic matter (SOM) levels after the application of weed control measures to a model crop. CO...control, mechanical removal of mustard; GBH...application of one glyphosate-based herbicide (Roundup PowerFlex, Roundup LB Plus, Touchdown Quattro), AI...application of one respective glyphosate active ingredient (potassium, isopropylamine, diammonium salt). Legacy effects from application of the same measures 4 months ago; short-term effects after new application on day 42 (treatment). Means ± SD, *n* = 5.

**Table 2.** Legacy effects of GBH or AI treatments over 150 days on springtails daily and cumulative activity in soil with either low or high soil organic matter (SOM). Springtail activity was assessed with pitfall traps during 4 sampling dates. Soil moisture, temperature, and electrical conductivity were measured at springtail sampling dates. Significances obtained from ANOVAs of generalized linear models. Significant effects are in bold.

| Parameters | Treatment | SOM | Treatment × SOM | Soil Temperature | Soil Moisture | Soil Electr. Conductivity |
|---|---|---|---|---|---|---|
| Springtail daily activity (ind. pot$^{-1}$) | **<0.001** | **0.018** | **0.037** | 0.617 | 0.209 | 0.973 |
| Springtail cumulative activity (ind. pot$^{-1}$) | **<0.001** | **0.003** | **0.014** | 0.774 | 0.778 | 0.673 |
| Soil moisture (% $v\,v^{-1}$) | **<0.001** | 0.219 | **<0.001** | **0.012** | - | **<0.001** |
| Soil temperature (°C) | 0.997 | 0.317 | 0.991 | - | **0.012** | **0.003** |
| Soil electrical conductivity (mS m$^{-1}$) | **0.049** | **<0.001** | 0.688 | **0.003** | **<0.001** | - |

The interactive effects between treatments and SOM varied among treatments. For the GBHs, springtail activity was significantly higher at high SOM for LB and TQ. For AIs or the control treatment, the effects of SOM were not clear (Figure 2; Table 3).

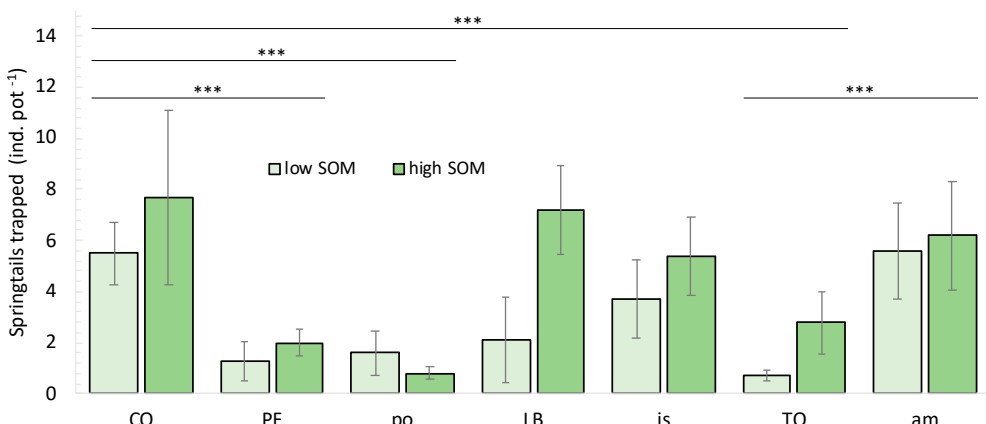

**Figure 2.** Legacy effects over 150 days on springtail activity in soil with low (3.0%) or high (4.1%) soil organic matter levels (SOM) where different weed control measures have been applied. CO...control, mechanical removal of mustard; Roundup PowerFlex (PF) or its AI glyphosate potassium salt (po); Roundup LB Plus (LB) or its AI isopropylamine salt (is); Touchdown Quattro (TQ) or its AI diammonium salt (am). Means ± SE of four sampling dates, *n* = 5. Significant differences between CO and individual GBHs or AIs, and between GBHs and associated AIs are indicated with asterisks; *** *p* < 0.001. See Table 3 for the statistical results of all mean comparisons.

**Table 3.** Legacy effects over 150 days of weed control measures with three GBHs, their respective AIs or mechanical removal of mustard (CO) on springtail activity. Pairwise comparisons of means after GLM analysis. GBHs: Roundup PowerFlex (PF), Roundup LB Plus (LB), Touchdown Quattro (TQ); AIs: glyphosate potassium salt (po), isopropylamine salt (is), diammonium salt (am). GBH and respective AIs are in bold. Significance codes: 0–0.001 = ***, 0.001–0.01 = **, 0.01–0.05 = *. *p* values are adjusted stepwise. Estimate = difference between the means; z value = estimate/standard error.

| Pairwise Comparison | Estimate | Standard Error | z Value | *p* Value | Significance |
|---|---|---|---|---|---|
| PF-CO | −1.45071 | 0.31956 | −4.540 | <0.001 | *** |
| po-CO | −1.73374 | 0.33725 | −5.141 | <0.001 | *** |
| LB-CO | −0.53461 | 0.28792 | −1.857 | 0.50493 | |
| is-CO | −0.43864 | 0.28045 | −1.564 | 0.7015 | |
| TQ-CO | −1.60131 | 0.33779 | −4.741 | <0.001 | *** |
| am-CO | −0.11703 | 0.27314 | −0.428 | 0.99952 | |
| **po-PF** | **−0.28303** | **0.37647** | **−0.752** | **0.98896** | |
| LB-PF | 0.916096 | 0.33370 | 2.745 | 0.08619 | |
| is-PF | 1.012065 | 0.32735 | 3.092 | 0.03227 | * |
| TQ-PF | −0.1506 | 0.37705 | −0.399 | 0.99968 | |
| am-PF | 1.33367 | 0.32130 | 4.151 | <0.001 | *** |
| LB-po | 1.199128 | 0.35061 | 3.420 | 0.01095 | * |
| is-po | 1.295096 | 0.34459 | 3.758 | 0.00331 | ** |
| TQ-po | 0.132431 | 0.39201 | 0.338 | 0.99988 | |
| am-po | 1.616702 | 0.33888 | 4.771 | <0.001 | *** |
| **is-LB** | **0.095968** | **0.29671** | **0.323** | **0.99991** | |
| TQ-LB | −1.0667 | 0.35117 | −3.038 | 0.03802 | * |
| am-LB | 0.417574 | 0.28990 | 1.440 | 0.77656 | |
| TQ-is | −1.16267 | 0.34514 | −3.369 | 0.01298 | * |
| am-is | 0.321606 | 0.28248 | 1.139 | 0.91467 | |
| **am-TQ** | **1.484271** | **0.33943** | **4.373** | **<0.001** | *** |

Looking at the individual treatments, springtail activity was affected differently by different GBHs or AIs (Figure 2). Among the GBHs, pots treated with Roundup LB Plus showed the highest activity, followed by those treated with Touchdown Quattro and Roundup PowerFlex. However, all GBH pots had lower activity than control pots, which was significant in the case of Roundup PowerFlex and Touchdown Quattro (Table 3). Legacy effects resulted in significant differences in activity between Touchdown Quattro and its AI diammonium salt (Table 4). Roundup Powerflex and its AI potassium salt, and Roundup LB Plus and its AI isopropylamine salt had similar effects on springtail activity (Figure 2).

**Table 4.** Short-term effects of GBH or AI treatments over 35 days on springtails daily and cumulative activity in soil with either low or high soil organic matter (SOM). Springtail activity was assessed with pitfall traps during 4 sampling dates. Soil moisture, temperature, and electrical conductivity were measured at springtail sampling dates. Significances obtained from ANOVAs of generalized linear models. Significant effects are in bold.

| Parameter | Treatment | SOM | Treatment × SOM | Soil Temperature | Soil Moisture | Soil Electr. Conductivity |
|---|---|---|---|---|---|---|
| Springtail daily activity (ind. pot$^{-1}$) | **<0.001** | 0.782 | **<0.001** | **0.011** | 0.691 | 0.792 |
| Springtail cumulative activity (ind. pot$^{-1}$) | **<0.001** | 0.402 | **<0.001** | 0.698 | 0.649 | **0.047** |
| Soil moisture (% $v\,v^{-1}$) | **<0.001** | **<0.001** | **<0.001** | 0.022 | - | **<0.001** |
| Soil temperature (°C) | 0.253 | 0.429 | 0.274 | - | **0.022** | 0.057 |
| Soil electr. cond. (mS m$^{-1}$) | 0.271 | **<0.001** | 0.552 | 0.057 | **<0.001** | - |

Legacy effects resulted in lower springtail activity in all AI pots compared to the control pots (Figure 2). TDR measurements of soil moisture, soil temperature, and soil electrical conductivity had no effect on springtail activity (Table 3).

Cumulative springtail activity was significantly affected by previous weed control treatment, SOM, and their interactions (Table 2). Pots with high SOM had higher cumulative activity across all treatments (5.6 ± 0.8 ind. pot$^{-1}$), compared to pots with low SOM content (3.5 ± 0.6 ind. pot$^{-1}$, Figure 3, Table 3).

Cumulative activity among GBHs, AIs, and controls was greater in high SOM pots. The largest difference between SOM was observed in GBH pots, with 1.6 ± 0.7 ind. pot$^{-1}$ at low SOM, and 4.9 ± 1.0 ind. pot$^{-1}$ at high SOM (Figure 3).

The mean soil moisture before treatment was 19.1 ± 0.2%. For the legacy effects of treatment as well as the treatment × SOM, the interaction significantly influenced the soil moisture, but not SOM alone (Table 2). However, soil temperature or electrical conductivity did not affect cumulative springtail activity.

There were no legacy effects on plant height. Plants had an average height of 9.8 ± 2.1 cm when treated with GBHs/AIs or hand-weeded.

*3.2. Short-Term Effects on Springtail Activity*

Daily springtail activity after treatment was 8.1 ± 0.5 individuals pot$^{-1}$, about twice that of the previous period investigating legacy effects. SOM had no effect on springtail activity (Figure 1). The mean number of springtails per pot across treatments was 8.5 ± 0.8 in low SOM soils compared to 7.7 ± 0.5 in high SOM soils.

Treatments significantly affected springtail activity (Table 4). Although SOM alone did not affect springtail activity, we found a significant treatment × SOM interaction (Table 4). Pots treated with GBHs and AIs had fewer springtails per sample on average than control pots (Figure 1). Individual GBHs and AIs had lower activity levels than the control in the low SOM soil (Figure 4).

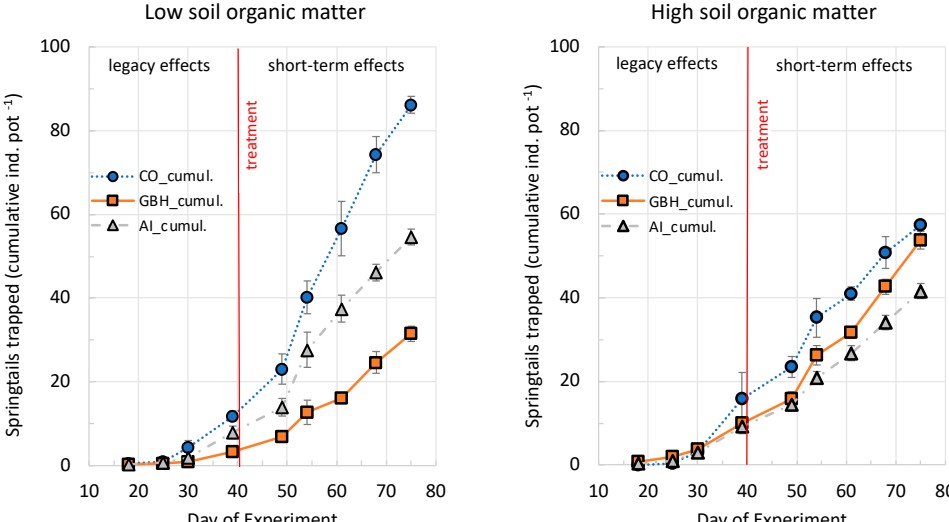

**Figure 3.** Cumulative springtail activity under low (3.0%) and high (4.1%) soil organic matter (SOM) levels after the application of weed control measures to a model crop. CO...control, mechanical removal of mustard; GBH...application of one glyphosate-based herbicide (Roundup PowerFlex, Roundup LB Plus, Touchdown Quattro), AI...application of one respective glyphosate active ingredient (potassium, isopropylamine, diammonium salt). Legacy effects from application of the same measures 150 days ago; short-term effects after new application on day 42 (treatment). Means ± SD, *n* = 5.

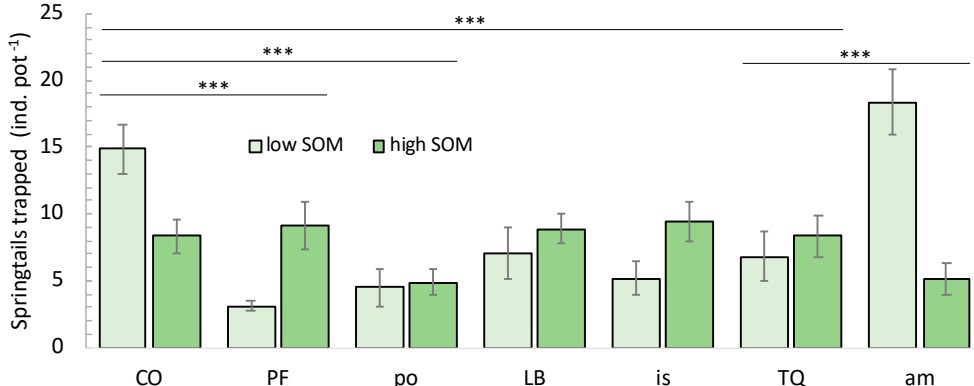

**Figure 4.** Short-term effects over 35 days on springtail activity in soil with low (3.0%) or high (4.1%) soil organic matter levels (SOM) where different weed control measures have been applied. CO...control, mechanical removal of mustard; Roundup PowerFlex (PF) or its AI glyphosate potassium salt (po); Roundup LB Plus (LB) or its AI isopropylamine salt (is); Touchdown Quattro (TQ) or its AI diammonium salt (am). Means ± SE of four sampling dates, *n* = 5. Significant differences between CO and individual GBHs or AIs, and between GBHs and associated AIs are indicated with asterisks; *** $p < 0.001$. See Table 5 for the statistical results of all mean comparisons.

**Table 5.** Short-term effects of weed control measures over 35 days with three GBHs, their respective AIs or mechanical removal of mustard (CO) on springtail activity. Pairwise comparisons of means after GLM analysis. GBHs: Roundup PowerFlex (PF), Roundup LB Plus (LB), Touchdown Quattro (TQ); AIs: glyphosate potassium salt (po), isopropylamine salt (is), diammonium salt (am). GBH and respective AIs are in bold. Significance codes: 0–0.001 = ***, 0.001–0.01 = **, 0.01–0.05 = *. *p* values are adjusted stepwise. Estimate = difference between the means; z value = estimate/standard error.

| Pairwise Comparison | Estimate | Standard Error | z Value | *p* Value | Significance |
|---|---|---|---|---|---|
| PF-CO | −1.51015 | 0.29680 | −5.088 | <0.001 | *** |
| po-CO | −1.66932 | 0.30368 | −5.497 | <0.001 | *** |
| LB-CO | −0.55884 | 0.26540 | −2.106 | 0.345 | |
| is-CO | −0.47747 | 0.25830 | −1.849 | 0.511 | |
| TQ-CO | −1.63288 | 0.30899 | −5.285 | <0.001 | *** |
| am-CO | −0.12794 | 0.25076 | −0.510 | 0.999 | |
| **po-PF** | **−0.15917** | **0.34323** | **−0.464** | **0.999** | |
| LB-PF | 0.95131 | 0.31068 | 3.062 | 0.035 | * |
| is-PF | 1.03268 | 0.30472 | 3.389 | 0.012 | * |
| TQ-PF | −0.12273 | 0.34799 | −0.353 | 0.999 | |
| am-PF | 1.38221 | 0.29862 | 4.629 | <0.001 | *** |
| LB-po | 1.11048 | 0.31722 | 3.501 | 0.008 | ** |
| is-po | 1.19185 | 0.31139 | 3.828 | 0.002 | ** |
| TQ-po | 0.03644 | 0.35378 | 0.103 | 1.000 | |
| am-po | 1.54138 | 0.30546 | 5.046 | <0.001 | *** |
| **is-LB** | **0.08137** | **0.27448** | **0.296** | **0.999** | |
| TQ-LB | −1.07404 | 0.32233 | −3.332 | 0.015 | * |
| am-LB | 0.43090 | 0.26751 | 1.611 | 0.672 | |
| TQ-is | −1.15541 | 0.31659 | −3.650 | 0.005 | ** |
| am-is | 0.34953 | 0.26047 | 1.342 | 0.830 | |
| **am-TQ** | **1.50494** | **0.31074** | **4.843** | **<0.001** | *** |

Looking at the individual treatments, all three GBHs had lower springtail activity than the control pots. Among the GBHs, SOM effects were most pronounced in PF, while no SOM effects were observed in LB and TQ pots (Figure 4). Among the AIs, potassium salt had the lowest springtail activity, which was significantly lower than that of the control group ($p < 0.001$). Isopropylamine salt had the second lowest activity of the AIs and diammonium salt the highest one, which was comparable to the control (Figure 4). None of the GBHs were significantly different from the corresponding AIs in springtail activity.

Electrical conductivity and soil moisture had no effect on springtail activity, while soil temperature had a significant negative effect on springtail activity. A decrease in soil temperature was associated with an increase in springtail activity.

Cumulative springtail activity was significantly affected by the GBH/AI application. SOM content alone had no effect on cumulative activity (Table 4). Mean cumulative activity was higher in pots with low SOM content than in pots with high SOM content. On average, $24.8 \pm 2.1$ ind. pot$^{-1}$ were captured at low SOM levels, and $21.9 \pm 1.4$ ind. pot$^{-1}$ were captured at high SOM levels.

Soil moisture was $21.7 \pm 0.3\%$ throughout the treatment period and ranged from 9.5 to 33.5%. There was a significant positive effect of electrical conductivity on cumulative springtail activity, but no significant relationship between soil moisture or soil temperature on cumulative activity was found (Table 5).

## 4. Discussion

### 4.1. Effects on Springtail Activity

We observed legacy effects on springtail activity from GBH/AI treatments 150 days earlier: three of six substances reduced springtail activity compared to the control group. The few studies that examined the effects of glyphosate on springtails over several months had mixed results. Two-year studies found a lower abundance of springtails after glyphosate

application [48,49]. Another study found no effect on springtail abundance after one year of glyphosate application [50].

We found higher surface activity at a higher SOM content. Even before the application of GBH/AI in the previous experiment, a higher springtail activity was found in soils with high SOM content [39]. Higher SOM content and reduced tillage in combination with pesticides as well as mineral fertilizers can stimulate springtails, but the management history of a site has an influence on the springtail community [51]. Springtail abundance and species diversity were studied for three consecutive years in fields treated with GBH and compared with mechanical weed control. Similar effects on springtail communities were found with mechanical weeding, although some species were more sensitive to the treatments than others [52]. Springtails correlated with weed cover, but other factors such as an abundance of predators (e.g., beetles) and rainfall also played a role. The species most affected by the treatments in their study were surface-active, litter-dwelling springtails [52], just like the species used in our study. Species-specific effects of three pesticides on four different springtail species have also been found by others [53,54]. Not surprisingly, springtail species generally differ in their sensitivity to chemicals [31,55], suggesting that studies should be expanded to include more species [56].

In the current study, springtail activity responded depending on SOM. At a low SOM, activity increased most for mechanical removal of mustard plants, second for the AIs, and third for GBHs. At a high SOM, activity was similar for mechanical removal of mustard and GBHs, while it was lower for AIs. This is in contrast to other studies, that found higher activities under GBH and linked this to a possible higher nutrient content of GBHs than AIs [39]. Different effects of the same substances suggest other factors influencing springtail activity in different experiments. For example, the current experiment was conducted in autumn and winter, whereas the previous experiment was conducted in the summer at much warmer ambient temperatures. In addition, different model plants were used, *Amaranthus retroflexus* in [39] vs. *Sinapis alba*, in the current experiment. Springtail activity was positively correlated with inter-row tillage and herbicide use in vineyards and has been attributed to negative effects on potential competitors and predators, stimulation of microorganisms, and/or increased nutrient inputs [44]. In addition to the effects on the activity of springtails, the study of physiological, biochemical, or ecological effects can shed light on the specific pathways through which GBHs or AIs affect springtails [25].

The presence of decaying plants favoured springtails, and the amount of weed detritus influenced the abundance of surface-active springtail species [57]. In our study, plant height did not differ between treatments, so we assume that the amount of plant residues was similar in all pots and did not explain activity patterns.

We found that Roundup Powerflex and its AI potassium salt had the lowest activity levels, and the AI potassium salt also had the lowest average activity after another application treatment. The AI potassium salt could be more harmful than the other AIs, and the GBHs differed considerably in their effects on springtails. Other studies also found that a GBH (Montana) reduced *Folsomia candida* proliferation at both 75% and 100% of the recommended application rate [58]. In contrast, others found significant avoidance behaviour of *F. candida* to one of four different GBHs [59]. The GBH containing only potassium salt as AI was avoided.

In a study with *F. candida*, partial synergistic effects on survival and reproduction rates were found for mixtures, as opposed to its single use of GBHs [38]. They concluded that ecological risk assessments should consider possible synergistic (higher toxicity in combination), antagonistic (lower toxicity in combination), and dose-dependent effects of the tested substances [38]. Considering the specific effects of the tested GBHs and AIs, synergistic or antagonistic effects could occur due to the composition of the GBH formulation.

The pitfall traps used in our experiment did not catch the springtail species *Sinella tenebricosa* originally introduced into the experimental pots, but a different species, *Sminthurinus niger*. This was also shown in another study [39] and suggests that the conditions in the experimental units were not suitable for this species, or that the pitfall trap was not the

appropriate method to collect it. Prior to our experiment, we stored the two soils in a warm greenhouse over the summer months (with temperatures up to 40 °C), homogenised the soils and let them dry out completely. We therefore expected that this procedure would have depleted and inactivated soil fauna, which would have led to similar populations at the beginning of the experiment. We refrained from defaunating the soils by freezing, heating, microwaving, or chemical treatment, as these methods also have side effects on various soil parameters [60,61].

### 4.2. Effects of Soil Organic Matter Content

We hypothesized that the surface activity of springtails would be increased in soil with high SOM content because springtails are promoted by high organic matter content [62]. High SOM content goes hand in hand with high soil microbial biomass, which is the main driver of glyphosate degradation [18]. Other soil parameters such as mineral content, aluminium and iron oxide could also affect the binding of glyphosate and have been shown to be important as well [34], but were not determined in the current study. Glyphosate degradation in high SOM soil might have been faster compared to low SOM soil due to the higher abundance of activity of microorganisms. Importantly, the high SOM soil had also higher P and K contents, so more nutrients were available to the soil biota.

Springtails feed on SOM [63] and soil microorganisms [55], suggesting that the high SOM soil provided a greater food supply than the others in our experiment. This was confirmed because, in five of seven treatment levels, springtail activity was higher in the high SOM soil. The opposite was true in the control pots and the pots were treated only with the AI diammonium salt. The results of a previous experiment support our findings. An increased surface activity of *S. niger* was observed in pots with high SOM content, with significant GBH × SOM and AI × SOM interactions [39].

However, soil texture and other soil characteristics could also be used to explain the fate and mobility of glyphosate and its metabolites in the two different soils [34]. The pH value was the same in both soils, but we do not know the nitrogen content, mineral composition, or soil texture. For example, glyphosate could have synergistic or antagonistic interactions with metals in the soil [38]. Low phosphorous content may slow glyphosate degradation in soil [64], suggesting slower degradation in low SOM soil, which has lower P content than high SOM soils.

When testing the effects of four different herbicides on the springtail community, glyphosate was found to result in a short-term increase in springtail abundance until the community reached stability 40 days after application [65]. The authors hypothesized that abundance initially increased after glyphosate application because springtails can use the compounds as food. This could also be the case in our study, as the GBH pots in soils with low SOM showed a significantly greater increase in activity than in the other pots. Since the increase was greater in the GBH pots than in the AI pots and least in the control pots, the results suggest that the GBH pots provided more nutrients or other activity-stimulating ingredients than the AIs [20].

## 5. Conclusions

We found short- and long-term effects of GBHs/AIs on springtail activity, even when applied five months earlier. Both GBHs and AIs had long-term decreasing effects compared to mechanical removal of mustard control group. We also found that GBH/AI interacted with SOM, highlighting the influence of soil properties on chemical effects. In a second application of GBHs and AIs after five months, GBHs and AIs stimulated springtail activity in combination with the low SOM soil. Our expectation that there would be a difference in the effects of GBHs and AIs was confirmed for one of three GBH-AI combinations; Touchdown Quattro pots had significantly lower activity than pots with the AI diammonium salt throughout the study. It therefore appears to be product-specific regarding whether GBHs influence the activity of springtails differently and more often than AIs alone. This has also been confirmed for soil macrofauna such as earthworms [20,66–68]. As the co-formulants

in GBHs are not disclosed, we do not know to what extent the investigated GBHs differed in their ingredients and which substance influenced the observed springtail activities. In general, GBHs tended to have stronger effects on springtail activity than AIs, suggesting that co-formulants are not inert but modify the non-target effects of GBHs. The current study was conducted under controlled conditions in the greenhouse and focused only on springtails. The main results could also be transferred to the field situation as we used field soil. Ultimately, however, only field experiments can shed light on the effects in the field when many soil organisms including microorganisms, macrofauna and plant roots, interact with each other, even if the increasing complexity of uncontrollable factors brings new challenges. In any case, the results highlight the need to conduct pesticide risk assessments under real field conditions, taking into account soil and climate variables [8,66]. Based on our results, we suggest that ecotoxicological risk assessments should include long-term toxicity testing of glyphosate-AI, GBH and co-formulants for multiple soil biota to incorporate aspects of protecting soil biodiversity and soil health.

**Supplementary Materials:** The following supporting information can be downloaded at: https://www.mdpi.com/article/10.3390/agriculture13122260/s1, Table S1: DATA_MS_Altmanninger_31Oct 2023.xlsx.

**Author Contributions:** Conceptualization, J.G.Z., A.S., E.G., E.T., M.M. and S.K.; methodology, A.A., V.B., J.G.Z. and E.G.; validation, J.G.Z., A.S., E.G., E.T., M.M., S.K. and B.S.; statistical analysis, A.A. and B.S.; investigation, A.A. and V.B.; resources, E.G., E.T., M.M. and S.K.; data curation, A.A., E.G. and J.G.Z.; writing—original draft preparation, A.A. and J.G.Z.; writing—review and editing, all authors; visualization, J.G.Z.; supervision, J.G.Z. and A.S.; project administration, A.S., M.M., E.G. and J.G.Z.; funding acquisition, A.S. and J.G.Z. All authors have read and agreed to the published version of the manuscript.

**Funding:** This research was funded by project no. 97öu3 of the action Austria-Hungary of the Osztrák-Magyar Akció Alapítvány (OMAA) and the Austrian Agency for International Cooperation in Education and Research (OED) granted to AS and JGZ. The funding body had no role in the design of the study and collection, analysis, and interpretation of data and in writing the manuscript.

**Institutional Review Board Statement:** Ethical review and approval were waived for this study because only invertebrates were studied.

**Data Availability Statement:** All raw data are provided in the Supplementary Material.

**Acknowledgments:** We are grateful to Yoko Muraoka, Ricarda Schmidt, Katy Renner, and Cornelia Scholz for various help during the experimental and writing phase, and Pascal Querner for verification of the Collembola species. Thanks to the BOKU Experimental Farm for providing the soil.

**Conflicts of Interest:** The authors declare no conflict of interest.

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
