# Peer review of "Glyphosate-Based Herbicide Formulations and Their Relevant Active Ingredients Affect Soil Springtails Even Five Months after Application"

_agriculture, doi:10.3390/agriculture13122260_

Round 1
Reviewer 1 Report
Comments and Suggestions for Authors
This mansucript provides original data on the effect of glyphosate-based herbicides on activity of selected species of springtails based on a greenhouse experiment. It is a well written paper with clear goals and formulated hypotheses. All the methods used are adequate to this kind of research, including proper statistics and GLM data modeling. The results are appropriately discussed, conclusions are thorough, showing also limitations of this study and outlining the further progress in revealing of the herbicides effects on soil fauna in natural conditions.
I have only two points to be corrected/resolved:
1) It is not clear which collembolan species was used in the experiments since in MM section Sinella tenebricosa is mentioned, but in abstract Sminthurinus niger.
2) The reader would benefit from a schematic figure of distribution of eppendorf pitfall traps in a pot. From the design description it is not clear how many traps were installed within one pot.
Reviewer 2 Report
Comments and Suggestions for Authors
Based on the information provided about the study investigating the effects of glyphosate-based herbicides (GBHs) and their respective glyphosate active ingredients (AIs) on springtails, here are some questions that should be considered. I believe these questions can help delve deeper into the specific findings and implications of the study, promoting a better understanding of the effects of glyphosate-based herbicides on springtails and soil ecosystems:
- What were the specific findings regarding the effects of different GBHs and their respective AIs on springtail activity in soils with low and high organic matter content?
- How do the effects of GBHs and glyphosate AIs compare to mechanical weeding in terms of their impact on springtail activity?
- What were the observed short-term effects of GBHs and AIs on springtail activity, and how long did these effects persist (up to five months after application)?
- Were there significant differences in the effects of GBHs and AIs on springtail activity, indicating variations in toxicity or other factors among the different formulations?
- How did the organic matter content of the soil (low vs. high) influence the effects of GBHs and AIs on springtail activity? Were the impacts more pronounced in one type of soil?
- Did the study provide any insights into the mechanisms by which GBHs and AIs affected springtails? For example, did it explore behavioral changes, reproductive effects, or other physiological responses?
- Were there any specific co-formulants or inert ingredients in the GBH formulations that were found to have significant effects on springtail activity?
- How do the findings of this greenhouse study relate to real-world agricultural practices and the potential impact of GBHs on springtails in field conditions?
- Were there any recommendations or implications provided by the study regarding the use of GBHs, disclosure of formulation ingredients, or the need for considering soil properties in risk assessments?
- What are the limitations of the study, and are there any avenues for further research to better understand the effects of GBHs and AIs on springtails and other soil organisms?
- The study focused on the epedaphic springtail species Sminthurinus niger. It is important to consider the generalizability of the findings to other springtail species or soil organisms. Different species may exhibit varying sensitivities to glyphosate and GBHs, and their ecological roles and behaviors may differ.
- Exploring the underlying mechanisms by which glyphosate and GBHs affect springtails can provide a deeper understanding of the observed impacts. Investigating potential physiological, biochemical, or ecological mechanisms can shed light on the specific pathways through which these herbicides influence springtail activity.
- Understanding the fate and transport of glyphosate and its breakdown products in the environment is crucial. Assessing the persistence, mobility, and potential accumulation of these compounds in soil can help evaluate their long-term effects on soil organisms, including springtails.
- While the greenhouse experiment provides valuable insights, field studies are essential to validate the findings in real-world agricultural settings. Field studies can account for factors such as weather conditions, soil heterogeneity, and interactions with other organisms, providing a more comprehensive understanding of the ecological impacts of glyphosate and GBHs.
- Considering the broader ecological context is important when assessing the effects of glyphosate on soil organisms. Understanding the interactions between springtails and other soil organisms, such as microorganisms, earthworms, or plant roots, can provide insights into the potential cascading effects of glyphosate on soil ecosystems.
- The study's implications for regulatory considerations and risk assessment frameworks should be explored. It raises questions about the adequacy of current guidelines and the need for incorporating long-term effects, and soil properties into risk assessments to ensure the protection of soil organisms and ecosystem health.
Comments on the Quality of English Language
Fair enough.
Reviewer 3 Report
Comments and Suggestions for Authors
Please refer to each comment stated in the edited PDF (attached). Point by point.
Below are major comments and edits
1. Title should be modified to be :Glyphosate -based herbicide formulations and their relevant active ingredients affect soil springtails five months after application
You did not use the pure glyphosate acid (technical material), rather you typically used active ingredient (they are dissimilar)
2. Line 108: The control check is UNWEEDED , UNTREATED ; Mechanical/ hand weeding is separate and different weed control methods
3. Line 42: please verify that?? AI can reach 75 % in some formulations
4.Line 133: Please explain how did you spray non-selective herbicide (glyphosate) on mustard plants???
5. Line 144:Control means UNTREATED pots .Mechanical weeding is considered weed control treatment (hoeing, hand removal)
6. Line 164: Please explain clearly how did you spray herbicide Ai and formulations?? spray volume, spraying equipment,...
7. Line 170-172: For meaningful comparison, the active ingredient % contained in the salts and commercial formulations of glyphosate should be the same with the AI. So the differential effects would be due to adjuvants and co-formulants
8. Line 197: Did you check for mortality of springtail?? or any other sublethal effects?? reproduction, behavior, developmental effects, enzyme activities???? Did you calculate LC50/ LD50,....
9. Line 431:Did you determine soil minerals; aluminum oxide and/ or iron oxide which are the major minerals that strongly bind with glyphosate forming glyphosate-metal chelation complex, causing high sorption force .Minerals are more influential than SOM in terms of glyphosate binding to soil.
Please refer to:
Kanissery R, Gairhe B, Kadyampakeni D, Batuman O, Alferez F. Glyphosate: Its Environmental Persistence and Impact on Crop Health and Nutrition. Plants (Basel). 2019 Nov 13;8(11):499. doi: 10.3390/plants8110499. PMID: 31766148; PMCID: PMC6918143.

Reviewer 4 Report
Comments and Suggestions for Authors
The authors investigated the short-term and long-term effects of different glyphosate herbicides and glyphosate (active ingredient) on the activity of springtails. The manuscript is well-written (references should be corrected) and the research design is complex but nicely executed, as well as the statistical analysis. The work is interesting and provides important, environmentally relevant information necessary for understanding the impact of glyphosate-based herbicides on soil organisms.
However, I recommend that the authors address certain issues that arose during the experiment.
From the experiment description, it is apparent that natural soil was utilized, and based on the results, it seems the soil was not pretreated (defaunated – frozen or heated to remove other soil organisms). Given that Sinella tenebricosa, presumably the target species, was not captured according to the results, did the authors initially assess the fauna present in the soils used for the experiments (low SOM and high SOM)? Were these faunal compositions similar? Were other organisms present in the soil used for the experiments? What could explain the absence of Sinella tenebricosa in your traps? Is it possible mortality occurred?
While I don't see an inherent issue in introducing Sinella tenebricosa and later opting for S. niger for statistical analysis, some clarification is needed. Why was this species introduced if the soil was not defaunated, especially if the aim was to evaluate the effects on different collembola species already present in the soil? The research question should be explicitly stated at the end of the Introduction section.
The validity of the experiment hinges on the similarity of soil fauna between the different soils (low SOM and high SOM). If there are significant differences in fauna composition, it poses a potential problem. Please include clarifications in the Materials and Methods section and discuss these aspects in the Results and Discussion sections. This will enhance the transparency and robustness of your study.
Figures -please add letters for statistical significance (compact letter display) on all figures, as it will be easier to follow the results.
Round 2
Reviewer 2 Report
Comments and Suggestions for Authors
Thanks for addressing all my comments.
Reviewer 3 Report
Comments and Suggestions for Authors
Please use mechanical removal of mustard instead of mechanical weed control.
It is confusing to use the word (weed) for mustard plant